# Antifungal Resistance and Genotyping of Clinical *Candida parapsilosis* Complex in Japan

**DOI:** 10.3390/jof10010004

**Published:** 2023-12-20

**Authors:** Hazim O. Khalifa, Akira Watanabe, Katsuhiko Kamei

**Affiliations:** 1Department of Veterinary Medicine, College of Agriculture and Veterinary Medicine, United Arab Emirates University, Al Ain P.O. Box 1555, United Arab Emirates; 2Medical Mycology Research Centre, Division of Clinical Research, Chiba University, Chiba 260-8673, Japan; fewata@faculty.chiba-u.jp (A.W.); chibakamei2@gmail.com (K.K.); 3Department of Pharmacology, Faculty of Veterinary Medicine, Kafrelsheikh University, Kafr El-Sheikh 33516, Egypt

**Keywords:** *Candida parapsilosis* complex, azole resistance, echinocandin resistance, *FKS1* hotspot mutations, microsatellite genotyping

## Abstract

Non-*albicans Candida* infections have recently gained worldwide attention due to their intrinsic resistance to different antifungal agents and the limited therapeutic options for treating them. Although the *Candida parapsilosis* complex is reported to be the second or third most prevalent *Candida* spp., little information is available on the prevalence of antifungal resistance along with genotyping of the *C. parapsilosis* complex. In this study, we aimed to evaluate the prevalence of antifungal resistance, the genetic basis of such resistance, and the genotyping of *C. parapsilosis* complex isolates that were recovered from hospitalized patients in Japan from 2005 to 2019. Our results indicated that, with the exception of one single *C. metapsilosis* isolate that was dose-dependently susceptible to fluconazole, all other isolates were susceptible or showed wild phenotypes to all tested antifungals, including azoles, echinocandins, amphotericin B, and flucytosine. Molecular analyses for azole and echinocandin resistance via evaluating *ERG11* mutation and *FKS1* hotspot one (HS1) and hotspot two (HS2) mutations, respectively, confirmed the phenotypic results. Genotyping of our isolates confirmed that they belong to 53 different but closely related genotypes, with a similarity percentage of up to 90%. Our results are of significant concern, since understanding the genetic basis of echinocandin resistance in the *C. parapsilosis* complex as well their genotyping is essential for directing targeted therapy, identifying probable infection sources, and developing strategies for overcoming epidemic spread.

## 1. Introduction

Pathogenic fungi have become more prevalent in recent decades, posing a rising threat to public health, especially considering the scarcity of antifungal medications available to treat invasive infections, as well as the emergence of antifungal resistance [1]. According to a recent estimate, fungal infection affects over a billion people and kills more than 1.5 million per year, which is similar to the outcomes of tuberculosis and is more than three-fold greater than the rates caused by malaria [2]. The vast majority of annual deaths due to fungal infection are initially attributed to *Candida* and *Aspergillus* infections, which cause a high economic burden for the health care system [3,4]. Among *Candida* infections, a recent concern has been directed to non-*albicans Candida* infections, owing to their intrinsically decreased susceptibility to commonly used antifungal drugs together with their increasing infection rates, and the development of their resistance to echinocandins and azole derivates [3,4,5,6,7].

*C. parapsilosis* is reported to be the second or third most prevalent *Candida* spp. in certain geographical regions, including Japan [5,8]. For instance, *C. parapsilosis* is the second major cause of candidemia in Japan [8], Spain [9], and Iran [10]. Furthermore, candidemia associated with *C. parapsilosis* has increased two-fold between 2008 and 2011 in North America, was responsible for 10 to 20% of all candidemia cases, and was associated with a wide range of clinical manifestations, including meningitis, endocarditis, vulvovaginitis, ocular infections, and urinary tract infections [11]. The problems of *C. parapsilosis* infections are complicated by their increased MIC valuesto the first-line antifungal therapy (echinocandin), as compared to *C. albicans* or *C. glabrata*, with differences at the species level [12]. Furthermore, recent reports indicate the emergence of fluconazole-resistant *C. parapsilosis* isolates, which has been associated with invasive infections [10,13]. Based on genomic analysis, the *C. parapsilosis* complex consists of three genetically distinct species: *C. parapsilosis sensu stricto*, *C. orthopsilosis*, and *C. metapsilosis*, which are phenotypically indiscernible from one another [10,11,12,13].

*C. parapsilosis* complex echinocandin resistance is exclusively attributed to the active mutations of *FKS1* gene hotspot regions (HS1, HS2) that encode the 1,3-β-D-glucan synthase complex enzyme [10,13]. *FKS1* hotspot mutations have been confirmed as a predisposing factor of therapeutic failure in candidemic patients and are basically related to prior echinocandin therapy [14]. Regarding azole resistance, two major mechanisms were reported in *C. parapsilosis*: (i) reduced azole accumulation caused by overexpression of the *CDR1*, *CDR2*, and *MDR1* genes, causing active efflux of drugs; and (ii) an active mutation in the drug target, the *ERG11* gene, which is associated with alterations in target protein structures, reductions in drug binding affinity, and a subsequently increased azole resistance [10,11,12,13].

To date, the prevalence of antifungal resistance, genetic mechanisms associated with resistance, and *C. parapsilosis* genotyping have never been tested in Japan. As far as we are aware, this is the first study to evaluate the epidemiology of antifungal resistance and genotyping of the *C. parapsilosis* complex recovered from clinical settings in Japan.

## 2. Materials and Methods

### 2.1. Candida parapsilosis Complex Isolates

In this study, a total of 79 clinical *C. parapsilosis* complex isolates recovered from 76 patients were tested, including 65 *C. parapsilosis* isolates recovered from 63 patients, 9 *C. metapsilosis* isolates recovered from 9 patients, and 5 *C. orthopsilosis* isolates recovered from 4 patients (Appendix A). The isolates were obtained from inpatients of different hospitals in 13 prefectures across Japan (Appendix A) during a 15-year period, from 2005 to 2019 (Appendix A). All of the isolates were provided through the National BioResource Project (NBRP), Japan “http://www.nbrp.jp/ (accessed on 19 December 2023)”. The study’s protocols and procedures were approved (approval number MMRC-REC 21-27) by the Ethical Committee of the Medical Mycology Research Center, Chiba University. Identification and confirmation of the isolates were performed via sequencing and analysis of the ITS1–5.8S rRNA–ITS2 DNA region, as previously described in [3,4,14].

### 2.2. Antifungal Susceptibility Testing

The antifungal susceptibility profiles of all of the isolates were determined by evaluating the minimum inhibitory concentrations (MICs) for the different antifungal agents fluconazole (FLC), voriconazole (VRC), itraconazole (ITC), and miconazole (MZ), as representatives of azoles, caspofungin (CAS) and micafungin (MFG), as representatives of echinocandins, and amphotericin B (AMB) and flucytosine (5FC), through broth microdilution assays according to CLSI document M27-Ed4, using Eiken dried yeast-like fungal DP plates EF-47 (Eiken Chemicals, Tokyo, Japan) [15]*. C. parapsilosis* ATCC 22019 and *C. krusei* ATCC 6258 were tested as quality control strains and the antifungal breakpoints were reported according to CLSI document M60 [16]. Resistances to FLC, CAS, and MFG were reported when the MIC values were ≥8 μg/mL, and were reported for VRC when the MIC value was ≥1 μg/mL [16]. The susceptibility profiles of ITC, AMB, and 5FC were recorded according to the epidemiological cutoff values (ECVs), and an isolate was reported as a non-wild type (non-WT) when the ECVs were >0.5, >2, and >0.5, respectively [17]. On the other hand, there are no established breakpoints or ECVs for MZ [16,17].

### 2.3. Genomic DNA Extraction

The genomic DNA of all isolates was extracted as previously described for *Candida* spp., with minor modifications [3,14]. Briefly, all isolates were grown on Sabouraud dextrose agar (SDA) for 24–48 h at 35 °C, followed by mixing and vigorous vertexing of 1 to 2 loopfuls of the yeast culture with 150  μL of lysis buffer consisting of 30 mM EDTA, 0.5% (*w*/*v*) sodium dodecyl sulfate, and 200 mM Tris-HCl (pH 8.0). After incubation at 100 °C for 20 min, the solution was mixed with 150  μL phenol–chloroform–isoamyl alcohol (25:24:1) and centrifugated at 13,000 rpm for 4 min. The clear supernatant was mixed with 300  μL of previously chilled 96% ethanol in a new Eppendorf tube. The solution was gently mixed and incubated in ice for 10–15 min, followed by centrifugation at 13,000 rpm for 15 min at 4 °C for DNA precipitation. After washing each DNA pellet with 500  μL of previously chilled 70% ethanol, the pellet was dried and suspended in 100–200 μL of sterile TE buffer or sterile distilled water, followed by preservation at −20 °C. Before the PCR experiments, the DNA template was prepared with a 10-fold dilution of DNA in sterile distilled water, and 1 μL of the resulting solution was used. 

### 2.4. Detection of ERG11 Mutations

PCR and DNA sequencing was performed to check for the presence of *ERG11* mutations in all *C. parapsilosis* and *C. orthopsilosis* isolates. For *C. parapsilosis ERG11* (*CpERG11*), NCBI accession number NW_023503279.1 for *C. parapsilosis* strain CDC317 was used for the design of primers and the *ERG11* sequence of *C. parapsilosis* ATCC 22019 was used as reference. For PCR and DNA sequencing of *ERG11*, four newly designed primers were used, and they are listed in Appendix A. For *C. orthopsilosis ERG11* (*CoERG11*), two previously published primers were used for the PCR experiments (Appendix A). Besides these primers, two other newly designed primers were used for the sequence of Co*ERG1* based on NCBI accession number MG601484.1 for *C. orthopsilosis* isolate Rome1 (Appendix A). Unfortunately, the *C. metapsilosis* ERG11 (CmERG11) sequence is not available in the database, hence the CmERG11 sequence was not investigated in this study.

### 2.5. Detection of FKS1 (HS1 and HS2) Mutations

GenBank accession numbers EU221325.1, XM_003867859.1, and EU350514.1 for *C. parapsilosis*, *C. orthopsilosis,* and *C. metapsilosis*, respectively, were used as a reference and for the primer design of the *FKS1* HS1 and HS2 regions. For the PCR reactions and DNA sequencing of both regions, four primers were designed and used for every species (Appendix A and Appendix A). 

### 2.6. Microsatellite Typing of C. parapsilosis Isolates

Genotyping of *C. parapsilosis* isolates was performed based on the microsatellite typing method using four loci designated as CP1, CP4, CP6, and B, composed of tandemly repetitive stretches of three nucleotides, which has previously been described to achieve a discriminatory power of 99.9% [18]. For exact and accurate allele size determination, the forward primers were fluorescently labeled with VIC dye for CP1, PET dye for CP4 loci, and FAM dye for CP6 and B5 loci (Appendix A). The alleles were designated according to their sizes (in base pairs) by using GeneScan™ 500 ROX™ Size Standard (Applied Biosystems, Warrington, UK) in the 35–500 nucleotide range and examined with PeakScanner (Thermo Fisher Scientific, Waltham, MA, USA). Based on the allele sizes of the four diploid loci for each isolate, a dendrogram was constructed by using BioNumerics v7.6 software (Applied Maths Inc., Austin, TX, USA) and a clustering method using the unweighted pair group method with average linkage (UPGMA) settings, as described previously [3,6].

### 2.7. Data Availability

The *C. parapsilosis ERG11* gene, *FKS1* HS1 region, and *FKS1* HS2 region sequences reported in this study have been deposited in GenBank under accession numbers OR536963 to OR537027, OR537028 to OR537092, and OR537093 to OR537157, respectively. The *C. orthopsilosis ERG11* gene, *FKS1* HS1 region, and *FKS1* HS2 region sequences reported in this study have been deposited in GenBank under accession numbers OR537158 to OR537162, OR537163 to OR537167, and OR537168 to OR537172, respectively.

## 3. Results

### 3.1. Clinical Features of the Isolates

The detailed clinical information of the isolates evaluated in this study is recorded in Appendix A. In total, 79 clinical isolates of the *C. parapsilosis* complex were isolates from 76 patients, and the median age of the 65 patients whose ages were known was 62 years. Among the patients, 61.8% (47/76) were male, 22.4% (17/76) were female, and the sexes of the remaining 15.8% (12/76) were unknown. The majority of the isolates were recovered from hospitalized patients in the Chiba prefecture (65.8%; 50/76), followed by the Tokyo prefecture (13.2%; 10/76), Tokushima prefecture (3.9%; 3/76), Osaka and Kyoto prefectures (2.6%; 2/76 each), and Fukuoka, Tochigi, Gunma, Akita, Aichi, Gifu, Saitama, and Kanagawa prefectures (1.3%; 1/76 each), and a single isolate was from an unconfirmed prefecture. The isolates were mainly recovered from blood (72.2%; 57/79), followed by those recovered from vascular catheters and corneas (6.3%; 5/79 each), otorrhea (3.8%; 3/79), and nails, urine catheters, abscesses, the liver, renal pelvis fluid, feces, pharyngeal fluid, pus, and unknown sources (1.3%; 1/79 each). Most of the isolates (32.9%; 26/76) were recovered from patients suffering from underlying diseases including neoplasms, diabetes mellitus, and hematologic malignancies, followed by: unknown illnesses (19.7%; 15/76); gastric disorders (9.2%; 7/76); blood and/or blood vessel-associated disorders (6.6%; 5/76); CNS disorders, congenital disorders, and corneal infections, each at 5.2% (4/76); genetic, immunity-related, and traffic accident-related disorders, each at 2.6% (2/76); and both gastric and CNS disorders, kidney disorders, cardiac disorders, nail infections, and pneumococcal sepsis, each at 1.3% (1/76). Fifteen patients were confirmed as being treated with antifungal drugs and five patients were confirmed as not receiving any antifungal therapy, while antifungal treatment of the other patients was unknown.

### 3.2. Antifungal Susceptibility Profiling

For azoles, only a single *C. metapsilosis* isolate was susceptible to FLC in a dose-dependent manner (MIC = 4 µg/mL); all other isolates were susceptible to FLC (MIC < 4 µg/mL), and all of the isolates were susceptible to VRC (MIC ≤ 0.5 µg/mL) and showed wild-type (WT) phenotypes for ITC (MIC ≤ 0.5 µg/mL) (Table 1 and Appendix A). For echinocandins, all of the isolates were susceptible to MFG and CAS (MIC < 4 µg/mL). Furthermore, all isolates showed WT phenotype for 5-FC (MIC ≤ 0.05 µg/mL) and AMB (MIC ≤ 2 µg/mL). AMB showed the highest geometric mean MIC value (0.92), followed by CAS (0.8), MFG (0.66), FLC (0.47), 5FC (0.12), MZ (0.08), ITC (0.04), and VRC (0.02) (Table 1).

### 3.3. Mutations in the ERG11 Gene and FKS1 HS Regions

For *C. parapsilosis*, all isolates harbored *ERG11* gene-synonymous mutations at T591C, and 33 isolates had missense mutations at R398I as compared to *C. parapsilosis* ATCC 22019 (Appendix A). For *C. orthopsilosis*, four isolates harbored *ERG11* gene-nonsynonymous mutations at Y13C and F420S, and one isolate harbored nonsynonymous mutations at Q211K, F420S, A421V, and V481I as compared to *C. orthopsilosis* isolate Rome1 (Appendix A). However, none of the isolates with *ERG11* missense mutations showed a higher MIC value for azoles. Furthermore, *C. metapsilosis* was not tested for the *ERG11* sequence. Checking the HS1 and HS2 regions of *FKS1* for all *C. parapsilosis* isolates, five *C. metapsilosis* isolates and all *C. orthopsilosis* isolates confirmed the absence of missense mutations. 

### 3.4. MLST Genotyping, Phylogeny, and Population Genetics 

The microsatellite typing method using four loci designated as CP1, CP4, B, and CP6 loci was performed. Since *C. parapsilosis* is a diploid species [18], one or two PCR fragments per locus were produced for each strain, and each fragment was allocated to an allele. When a strain produced two PCR products, it was classified as heterozygous, whereas strains that produced only one amplification product were categorized as homozygous. Our analysis of the 63 isolates showed that all microsatellite loci were exhibiting between 15 and 30 alleles and were from 16 to 32 different genotypes (Table 2). The size ranges (bp) of the CP1, CP4, B, and CP6 alleles were 216–269, 253–479, 116–197, and 213–328, respectively (Table 2). The microsatellite genotyping using a panel of four loci markers identified 53 different genotypes (Appendix A, Figure 1), of which 50 were observed only once. Three genotypes, numbered one to three, were found multiple times, and they were identified from four, three, and two isolates, respectively, from nine different patients (Appendix A). The remaining 50 genotypes involved only one patient each, with four isolates being isolated from two different patients (two isolates each). Using the clustering approach and BioNumerics software version 7.6, phylogenetic analyses of the isolates were carried out in order to ascertain the links between the identified genotypes. Our results confirmed a close relationship between all genotypes, with a similarity percentage of up to 90% (Figure 1).

## 4. Discussion

Recently, special attention has been paid to non-*albicans Candida* (NAC) species infections, with particular interest in the *C. parapsilosis* infection, owing to it being reported as a major cause of candidemia in different countries [8,9,10,11]. The progressive increase in the rates of antifungal resistance in most *candida* infections, and in the *C. parapsilosis* complex in particular [13], along with the narrowing therapeutic options [7], emphasizes the importance of studying the prevalence of antifungal resistance, as well as their genotyping. 

In accordance with prior publications regarding other *Candida* spp. [3,4,6], our findings showed that *C. parapsilosis* infections are typically seen in elderly individuals and patients with underlying illnesses. Furthermore, previous reports have confirmed that, throughout the world, *Candida* species continue to be the leading cause of opportunistic infections, primarily affecting patients over 65 years old [19]. The propensity of *C. parapsilosis* to build a biofilm on catheters and other implanted devices makes it an exogenous pathogen that is primarily found on skin surfaces as opposed to mucosal surfaces. In nursing homes and hospitals, it is transmitted via hand contamination. The fact that elderly patients frequently get at-home health care with indwelling catheter use owing to various chronic conditions is consistent with the observation that most *C. parapsilosis* infections in our study are identified in elderly patients [19]. However, contrary to the statewide data reported by Pfaller et al. that suggest *C. parapsilosis*, a member of the NAC species, is responsible for the majority of invasive candidiasis cases in children (of nine years old) and neonates in North America [20], only 17% of the patients in this study were children. Furthermore, *C. parapsilosis* was one of the major NAC species responsible for neonatal candidiasis in different countries including Canada, the UK, and Norway [5].

Our results confirm the absence of azole and echinocandin resistance among the tested *C. parapsilosis* complex isolates. The low worldwide level of azole and echinocandin resistance in *C. parapsilosis* has also recently been confirmed by different studies [5,21,22,23,24,25]. For instance, surveys of fluconazole and itraconazole resistance among isolate collections revealed resistance rates ranging from 0 to 4.6%, and from 1.5 to 4%, respectively [5]. Furthermore, globally, the fluconazole resistance rate ranged between 2 and 5% among *C. parapsilosis* isolates [21,22], and fluconazole resistance was reported in 3.4% of 6023 examined isolates in a recent review [23]. Notably, 33 *C. parapsilosis* azole-susceptible isolates had *ERG11* missense mutations at R398I. Previous reports have confirmed the lesser role of R398I in azole resistance, as it was recently identified in fluconazole-susceptible *C. parapsilosis* isolates; and even when R398I was identified in resistant isolates, it was accompanied by other missense mutations such as *Tac1* L877P, *Tac1* L877P and *Mrr1* P250S, *Tac1* L877P and *Mrr1* S1081P, or *Tac1* L877P and *Mrr1* P295R [24]. Furthermore, our results confirm the absence of *ERG11* Y132F variants in Japan. On the other hand, azole-resistant outbreaks of *C. parapsilosis* associated with the Y132F substitution have been recently identified in different countries including South Korea [24], China [25], Mexico [26], Turkey [27], and Brazil [28]. However, conducting other large-scale nationwide studies is essential to monitor the prevalence of such important resistance mechanisms in Japan.

Also, in accordance with our findings regarding echinocandin resistance, in a prospectively collected series of *C. parapsilosis* isolates, only 0.6% were resistant to echinocandins [29], and a very recent study in China confirmed their very low level of resistance (0.03%) to echinocandins [30]. Our results verify that the MIC geometric means of both examined echinocandins (CAS and MFG) do not significantly differ from one other. Other studies have verified that caspofungin outperforms both micafungin and anidulafungin in terms of in vitro activity against *C. parapsilosis* isolates [30], which is consistent with global surveillance program reports [22,29,30]. The susceptibility of *Candida* species to echinocandins varies; among the three echinocandins, *C. albicans*, *C. tropicalis*, *C. glabrata*, and *C. lusitaniae* were generally most sensitive to micafungin, while *C. krusei* and *C. pelliculasa* were most vulnerable to anidulafungin [30]. Our results showed a close relationship between the MIC results and the results from the genetic analysis, as all of the isolates showed an absence of *FKS1*-HS missense mutations, which are responsible for echinocandin resistance. Our findings supporting the reliability of azole and echinocandin MIC values obtained via CLSI and EUCAST methods to evaluate the resistance in *C. parapsilosis* complex, which is unlike other *Candida* species such as *C. glabrata* [3] and *C. krusei* [4], especially regarding echinocandin resistance. With both *C. glabrata* and *C. krusei*, we have to depend on *FKS1* HS mutation rather than MIC (especially CAS) results to determine echinocandin-resistant isolates. 

*Candida* genotyping has a significant role in the detection of emerging clones and the identification of relationships between certain genotypes and virulence traits, mortality rates, and gene polymorphisms, along with in investigating the potential source of infection [3,4]. Microsatellite genotyping, which has a greater discriminative strength than other techniques like DiversiLab typing, was the method we used in this investigation [12,18]. Although microsatellite genotyping characterized that our isolates are classified into 53 different genotypes, phylogenetic analysis of the isolates confirmed the close relationship between all of the genotypes, with a similarity percentage up to 90%. As far as we know, this is the first report to confirm this close relationship between Japanese clinical *C. parapsilosis* isolates. The diversity of the genotypes detected in this study points to the possibility of numerous causes contributing to the occurrence of *C. parapsilosis* infections in Japan. In line with our findings, two recent studies in Brazil identified different *C. parapsilosis* genotypes among pediatric patients [31,32], but their results also confirmed whether these genotypes are phylogenetically related or not. Moreover, highly related genotypes have caused outbreaks of *C. parapsilosis* candidemia in neonatal intensive care units in the USA [33]. Furthermore, other studies have also documented the occurrence of clonal complexes of closely related genotypes as a result of microevolution caused by the inherent instability of microsatellite loci [34]. 

## 5. Conclusions

In conclusion, our findings confirm the absence of antifungal resistance among clinical and *C. parapsilosis* complex isolates recovered in Japan. Our phenotypic susceptibility results were supported by genetic examination, as all of the isolates showed the absence of the missense mutations responsible for azole and echinocandin resistance. For the first time, microsatellite genotyping and phylogenetic analysis has confirmed that different, closely related genotypes are responsible for *C. parapsilosis* infections in Japan. 

## Figures and Tables

**Figure 1 jof-10-00004-f001:**
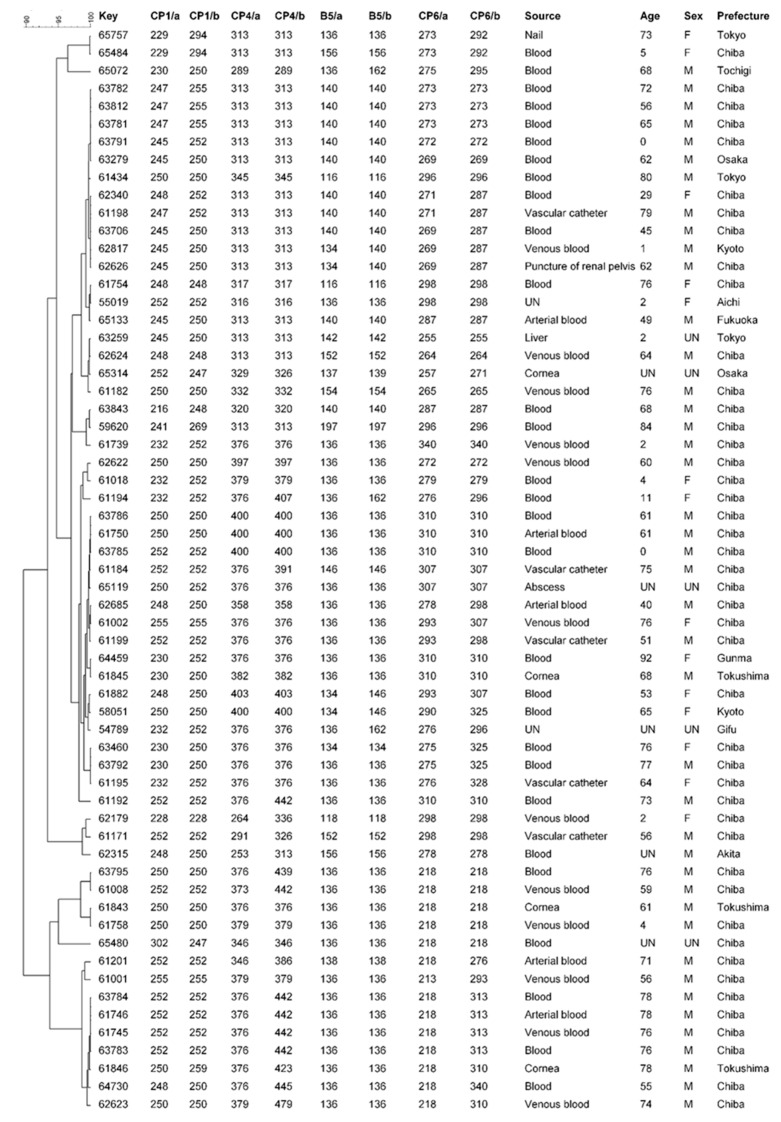
UPGMA dendrogram showing the similarities among 61 *C. parapsilosis* isolates (a single isolate for each patient) based on the microsatellite typing method using four loci designated as CP1, CP4, B, and CP6. Two isolates (IFM65553 and IFM64439) were not tested due to failure in the analysis of the CP4 segment despite several trials. Abbreviations: M, male; F, female; UN, unknown.

**Table 1 jof-10-00004-t001:** Summary of antifungal susceptibility profiling of *C. parapsilosis* complex isolates.

Drug	No. of Isolates at Each Determined MIC Value (µg/mL)									MIC Range (µg/mL)	GM ^a^ MIC (µg/mL)	MIC (µg/mL) of Quality Control Strains:	
	≤0.015	0.03	0.06	0.12	0.25	0.5	1	2	4			*C. parapsilosis* ATCC 22019	*C. krusei* ATCC 6258
MFG			1	1	13	14	50			0.06–1	0.66	0.5	0.12
CAS					1	23	55			0.25–1	0.8	1	0.25
AMB						10	69			0.5–1	0.92	0.5	1
5FC				79						0.12	0.12	≤0.12	4
FLC				1	24	42	6	5	1	0.12–4	0.47	1	16
ITC	5	38	26	10						0.015–0.12	0.04	0.06	0.12
VRC	59	18	2							0.015–0.06	0.02	0.03	0.12
MZ		16	27	22	14					0.03–0.25	0.08	0.12	0.25

^a^ GM, geometric mean. Abbreviations: MFG, micafungin; CAS, caspofungin; AMB, amphotericin B; 5FC, flucytosine; FLC, fluconazole; ITC, itraconazole; VRC, voriconazole; MZ, miconazole.

**Table 2 jof-10-00004-t002:** Characteristics of microsatellite loci for C. parapsilosis isolates.

Loci	Size Range (bp)	No. of Alleles	No. of Genotype
CP1	216–269	16	20
CP4	253–479	30	28
B5	116–197	15	16
CP6	213–328	27	32

## Data Availability

Data are contained within the article and Appendix A.

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
