# Peer review of "Antifungal Resistance and Genotyping of Clinical Candida parapsilosis Complex in Japan"

_jof, 2023, doi:10.3390/jof10010004_

Round 1

Reviewer 1 Report

Comments and Suggestions for Authors

This paper reports a straightforward and well written analysis of Candida parapsilosis isolates from Japan. Almost 80 isolates were evaluated by measuring resistance to antifungal drugs, and by sequencing known targets of antifungal drugs (e.g. ERG11). The authors showed that drug resistance levels were low, and they did not identify any variants known to be associated with drug resistance. The main contribution of the study is that it adds to worldwide data on the prevalence of drug resistance in C. parapsilosis.

I have only minor comments.

1. The authors note that all isolates are phylogenetically similar. However. C. parapsilosis isolates in general have very low diversity. The authors should clarify whether the Japanese isolates are more similar to each other that they are to isolates from other geographical areas. 90% similarity is lower than expected, so other MLST analyses should be discussed. It can be more useful to use cluster diagrams to show relationships, e.g using software like Bionumerics. An example applying this to MLST type data can be see in DOI: 10.1093/mmy/myac040.

2. It is somewhat surprising that the azole resistance levels are so low in recent isolates. There have been multiple reports of outbreaks associated with isolates with Y132F variants, e.g. in Turkey, US, Spain, Brazil. The authors should discuss these outbreaks in more detail. Their results suggest that Y132F is not yet present in Japan.

3. Please clarify the text around line 300 "In line with our findings, two recent studies in Brazil identified different C. parapsilosis genotypes among pediatric patients, but those results also confirmed whether these genotypes are phylogenetically related or not". Do you mean that they did not confirm phylogenetic realtionships?

Author Response

Comments and Suggestions for Authors

This paper reports a straightforward and well written analysis of Candida parapsilosis isolates from Japan. Almost 80 isolates were evaluated by measuring resistance to antifungal drugs, and by sequencing known targets of antifungal drugs (e.g. ERG11). The authors showed that drug resistance levels were low, and they did not identify any variants known to be associated with drug resistance. The main contribution of the study is that it adds to worldwide data on the prevalence of drug resistance in C. parapsilosis.

Response: Thank you for your comment and we appreciate your time in the revision of our manuscript.

I have only minor comments.

  1. The authors note that all isolates are phylogenetically similar. However. C. parapsilosis isolates in general have very low diversity. The authors should clarify whether the Japanese isolates are more similar to each other that they are to isolates from other geographical areas. 90% similarity is lower than expected, so other MLST analyses should be discussed. It can be more useful to use cluster diagrams to show relationships, e.g using software like Bionumerics. An example applying this to MLST type data can be see in DOI: 10.1093/mmy/myac040.

Response: Thank you for your comment. We have already tested the phylogenetic analysis using Bionumerics (L140-L143, and Figure 1). Furthermore, we checked the manuscript you recommended (DOI: 10.1093/mmy/myac040) and we found that the authors used multi-dye multiplex PCR using the same markers as we used (CP1, CP4a, CP6, B) and they didn’t use MLST for C. parapsilosis. MLST is not yet generated for C. parapsilosis, and microsatellite typing is the most commonly used and standardized method with achieving a discriminatory power of 99.9% (doi: 10.1128/JCM.02151-09.). On the other hand, in our previous research, we used MLST genotyping in C. glabrata (doi: 10.1128/aac.00783-20), C. tropicalis (doi: 10.1016/j.cmi.2021.10.004), and C. Krusei (doi: 10.1128/AAC.01856-21) as MLST is the standardised method for these Candida spp.

  1. It is somewhat surprising that the azole resistance levels are so low in recent isolates. There have been multiple reports of outbreaks associated with isolates with Y132F variants, e.g. in Turkey, US, Spain, Brazil. The authors should discuss these outbreaks in more detail. Their results suggest that Y132F is not yet present in Japan.

Response: Thank you for your comment. We completely agree with you, and therefore, we improved our discussion in the revised manuscript by highlighting this important point (L279-L284 and references 24-28).

  1. Please clarify the text around line 300 "In line with our findings, two recent studies in Brazil identified different C. parapsilosis genotypes among pediatric patients, but those results also confirmed whether these genotypes are phylogenetically related or not". Do you mean that they did not confirm phylogenetic realtionships?

Response: Thank you for your comment. The authors performed the phylogenetic analysis but they did not discuss if the isolates are closely related or not (similarity %). For example, in the manuscript described by Rodrigues et al, 2022 (doi: 10.3390/jof8121280), the authors confirmed that the isolates belonging to 38 different allelic profiles (genotypes) and three isolates (7.5%) shared an identical allelic profile (endemic clone). However, the similarity % of other isolates was not clearly described.

Reviewer 2 Report

Comments and Suggestions for Authors

The manuscript is relevant within the area it fits into. There are some points that need to be appreciated by the authors, corrected or explained.

Below are the points I highlight:

1 - In the title, correct the word "Candid";

2 - In several places "spp" is written, when the correct one would be "spp.";

3 - The introduction lacked information on which species make up the "Candida parapsilosis complex". It is important to mention briefly.

4 - At some points in the text it says "parapsilosi" when the correct word is "parapsilosis".

5 - Why in item "2.4. Detection of ERG11 mutations" do the authors not mention C. metapsilosis? In the supplementary material they mention, and in item 2.5 this species is also mentioned;

6 - In item "2.9. Data availability" the species C. metapsilosis is also not mentioned;

7 - Table 1 needs adjustments and formatting. The second column is very difficult to relate MIC to number of isolates;

8 - In the item "3.3 Mutations in the ERG11 gene and FKS1 HS regions" the species Candida metapsilosis was not mentioned;

9 - In figure 1, the acronym UN must be mentioned in the footer, with its meaning;

10 - The item "Institutional Review Board Statement:" on page 9 must be adapted to the authors or removed;

11 - In the "Supplementary Table S1 Characterization of clinical C. parapsilosis complex isolates recovered in this study", the authors need to correct the meaning of the acronym UN, which is wrong.

These are my considerations.

Author Response

Comments and Suggestions for Authors

The manuscript is relevant within the area it fits into. There are some points that need to be appreciated by the authors, corrected or explained.

Below are the points I highlight:

1 - In the title, correct the word "Candid";

Response: Thank you for your comment. We have already corrected it.

2 - In several places "spp" is written, when the correct one would be "spp.";

Response: Thank you for your comment. We have already corrected it throughout the manuscript (L15, L44).

3 - The introduction lacked information on which species make up the "Candida parapsilosis complex". It is important to mention briefly.

Response: Thank you for your comment. We have already added more information about species that make up the "Candida parapsilosis complex (L54-56).

4 - At some points in the text it says "parapsilosi" when the correct word is "parapsilosis".

Response: Thank you for your comment. We have corrected it (L69).

5 - Why in item "2.4. Detection of ERG11 mutations" do the authors not mention C. metapsilosis? In the supplementary material they mention, and in item 2.5 this species is also mentioned;

Response: Thank you so much for your comment. Unfortunately, the sequence of C. metapsilosis ERG11 is not available in the database or any publications despite several checks, and therefore, we can't design primers for PCR detection and sequence of C. metapsilosis ERG11. We have already mentioned this information in the revised manuscript (L122-L124).

 On the other hand, ERG11 sequence of C. parapsilosis and C. orthopsilosis is available and therefore we easily design primers for the detection and sequences.  However, we can detect the C. metapsilosis FKS1 (HS1 and HS2) by PCR and sequencing as it available in the database (GenBank accession number EU350514.1), for this reason, it was mentioned in item 2.5 and in the supplementary material related to FKS1 gene.

6 - In item "2.9. Data availability" the species C. metapsilosis is also not mentioned;

Response: Thank you so much for your comment. As I mentioned in the previous response, the sequence of C. metapsilosis ERG11 is not available and we did not check this gene by PCR and sequencing. For Candida metapsilosis FKS1, five isolates were checked by PCR and sequencing (revised manuscript L194-L196), and the full sequence of these five isolates will be fully described in our future research related to in vitro induction of echinocandin resistance in C. parapsilosis complex (under revision).

7 - Table 1 needs adjustments and formatting. The second column is very difficult to relate MIC to number of isolates;

Response: Thank you so much for your comment. We have already modified the second column by excluding the MICs more than 4 (which is not detected in this study) to exclude any conflict. Regarding the formatting, this format is required by the journal. Furthermore, the same table style was also used in our previous research on C. glabrata (doi: 10.1128/aac.00783-20), C. tropicalis (doi: 10.1016/j.cmi.2021.10.004), and C. Krusei (doi: 10.1128/AAC.01856-21).

8 - In the item "3.3 Mutations in the ERG11 gene and FKS1 HS regions" the species Candida metapsilosis was not mentioned;

Response: Thank you so much for your comment. As I mentioned in my previous responses (5 and 6), the C. metapsilosis ERG11 (CmERG11) sequence is not available in the database, hence the CmERG11 sequence was not investigated in this study (revised manuscript L122-L124, and L193-L194). For Candida metapsilosis FKS1, five isolates were checked by PCR and sequencing (revised manuscript L194-L195).

9 - In figure 1, the acronym UN must be mentioned in the footer, with its meaning;

Response: Thank you so much for your comment. The abbreviations used in Fig 1 were included in the Fig 1 legends (L242).

10 - The item "Institutional Review Board Statement:" on page 9 must be adapted to the authors or removed;

Response: Thank you so much for your comment. The "Institutional Review Board Statement:" was corrected in the revised version (L347-L349).

11 - In the "Supplementary Table S1 Characterization of clinical C. parapsilosis complex isolates recovered in this study", the authors need to correct the meaning of the acronym UN, which is wrong.

Response: Thank you so much for your comment. The acronym UN in the Supplementary Table S1 has been corrected in the revised version.

These are my considerations
